# Green Accelerated Hoeffding Tree

## ABSTRACT

State-of-the-art machine learning solutions mainly focus on creating highly accurate models without constraints on hardware resources. Stream mining algorithms are designed to run on resource-constrained devices, thus a focus on low power and energy and memory-efficient is essential. The Hoeffding tree algorithm is able to create energy-efficient models, but at the cost of less accurate trees in comparison to their ensembles counterpart. Ensembles of Hoeffding trees, on the other hand, create a highly accurate forest of trees but consume five times more energy on average. An extension that tried to obtain similar results to ensembles of Hoeffding trees was the Extremely Fast Decision Tree (EFDT). This paper presents the Green Accelerated Hoeffding Tree (GAHT) algorithm, an extension of the EFDT algorithm with a lower energy and memory footprint and the same (or higher for some datasets) accuracy levels. GAHT grows the tree setting individual splitting criteria for each node, based on the distribution of the number of instances over each particular leaf. The results show that GAHT is able to achieve the same competitive accuracy results compared to EFDT and ensembles of Hoeffding trees while reducing the energy consumption up to 70%.

## CCS CONCEPTS

• **Computing methodologies** → **Online learning settings**; **Classification and regression trees**; *Supervised learning by classification*; Ensemble methods; • **Hardware** → Impact on the environment; • **Computer systems organization** → Embedded systems.

## KEYWORDS

hoeffding trees, streaming algorithms, energy efficiency

**ACM Reference Format:**
Anonymous Author(s). 2020. Green Accelerated Hoeffding Tree. In *Proceedings of ACM Conference (Conference'17)*. ACM, New York, NY, USA, 8 pages. https://doi.org/10.1145/nnnnnnn.nnnnnnn

## 1 INTRODUCTION

Machine learning is advancing at a fast pace with the current state-of-the-art research. While the majority of solutions focus on obtaining as high accurate models as possible without evaluating the energy impact, during the past years some research lines have emerged that focus on developing low power models [15, 33, 35].

Data stream mining is an area in data mining where the inference and training of the algorithm are performed in real-time (or pseudo-real-time), building and updating the models incrementally. One of the key requirements is that only the statistics of the data are stored in the model. These models usually incur low computational and power demands, since their goal to run at the edge and on embedded devices. The Hoeffding Tree (HT) algorithm [9] is a pioneering algorithm for building decision trees for data streams with theoretical guarantees. Current research suggests that still the main focus of Hoeffding Tree extensions is to obtain a higher predictive accuracy compared to their predecessor [16]. This is the case of ensembles of Hoeffding trees [28–30] and the Extremely Fast Decision Tree (EFDT) [26]. These algorithms build more accurate trees, but at a high energy cost.

This paper presents the Green Accelerated Hoeffding Tree, an energy-efficient extension of the EFDT algorithm. GAHT sets individual splitting criteria for every leaf so that the algorithm can grow more organically depending on the distribution of the number of instances over the leaves of the tree. Distributing the growth of the tree between the different nodes allows for the algorithm to spend more energy on the part of the tree that needs to grow faster (increasing accuracy), and save energy on the other branches. GAHT has been empirically evaluated in terms of accuracy (percentage of correctly classified instances) and energy consumption against four baseline algorithms: HT [9], EFDT [26], OzaBag [28] (Online Bagging), and OzaBoost (Online Boosting) [28]. While HT consumes less energy than the rest of the algorithms, it also performs poorly in terms of accuracy—ranking as the worst algorithm in terms of accuracy for the majority of datasets. GAHT achieves better accuracy than EFDT and OzaBag. Although OzaBoost achieves 1% higher accuracy than GAHT on average, a statistical test on the ranks of the algorithms shows that there is no statistical difference in accuracy between such algorithms. However, the statistical test on the ranks between the energy consumption of the algorithms shows that GAHT consumes significantly less energy than EFDT, OzaBag, and OzaBoost. On the other hand, the difference in ranks in energy consumption between GAHT and HT is not statistically significant. On average GAHT consumes 27% less energy than EFDT, 67% less energy than OzaBag, and 72% less energy than OzaBoost.

Finally, we have evaluated the memory used by the models. GAHT outputs smaller trees than EFDT and the ensembles in the majority of the cases, outputting lower energy even for those cases when GAHT trees are bigger. These trees take up between 1MB to 30MB, which is in the range of the mobile nets, convolutional neural networks highly optimized for embedded devices [20].

The rest of the paper is organized as follows. Background explanations on HT, EFDT, OzaBag, and OzaBoost are given in Section 2. Related works to our are presented in Section 3. The details behind GAHT, together with converge and energy complexity analyses are presented in Section 4. Experiments and results are presented in Sections 5 and 6 respectively. Finally, conclusions and future lines of work are presented in Section 7.

## 2 PRELIMINARIES

This section explains in detail the Hoeffding Tree, EFDT, Online Bagging (OzaBag in the experiments), and Online Boosting (Oza-Boost in the experiments) while GAHT is explained in Section 4.

### 2.1 Hoeffding Tree

The Hoeffding algorithm (HT), proposed by [9], was the first algorithm that was able to mine from infinite streams of data, requiring low computational power.

The algorithm creates a decision tree in real-time obtaining similar performance as offline decision trees, with theoretical guarantees. This is achieved by saving the necessary information (statistics) from the different instances observed at each node. Once $nmin$ instances are observed at a leaf, the algorithm calculates the information gain (entropy) for each attribute, using the aforementioned statistics. If $\Delta \overline{G} > \epsilon$, i.e. the difference between the information gain from the best and the second-best attribute ($\Delta \overline{G}$) is higher than the Hoeffding Bound [19]($\epsilon$), then a split occurs, and the leaf substituted by the node with the best attribute. The Hoeffding bound($\epsilon$) is defined as:

$$\epsilon = \sqrt{\frac{R^2 \ln(1/\delta)}{2n}} \tag{1}$$

and it states that the split on the best attribute having observed $n$ number of examples, will be the same as if the algorithm had observed an infinite number of examples, with probability $1 - \delta$. On the other hand, if $\Delta \overline{G} < \epsilon < \tau$, a tie occurs. This happens when the two top attributes are equally good, thus the tree splits on any of those.

### 2.2 Extremely Fast Decision Tree

The EFDT is the implementation of the Hoeffding Anytime Tree, which was recently published at KDD [26]. The EFDT is able to create a Hoeffding tree with higher predictive performance than the Hoeffding Tree. This is achieved by having a less restrictive split criterion, allowing the tree to grow faster; and by reevaluating the already split nodes to try to resemble more the asymptotic batch decision tree. EFDT evaluates if the information gain of the best attribute is higher, by a difference of $\epsilon$, than the information gain of not splitting on the leaf. If that occurs, there is a split on the best attribute. While the EFDT can output significantly higher accuracy than standard Hoeffding trees, it does this at a higher cost of energy consumption.

### 2.3 Online Bagging

Bagging consists of creating $M$ different models that are trained on different samples of the data. Each sample is chosen randomly with replacement, i.e. the same sample can potentially be used to train more than one of the $M$ models. The final model makes a prediction by taking the majority voted class between the $M$ models.

Online Bagging [28, 29] is the extension of the bagging technique for online and streaming scenarios. The main challenge is that is not possible to draw examples randomly with replacement since the examples are only read once. To address that challenge, this algorithm reads each instance assigning it a specific weight following a Poisson distribution with $\lambda = 1$, for each model. After each instance is read, each model will be trained on the new instance $k$ times, where $k = Poisson(1)$. The rationale behind using a Poisson distribution lies in the property that a *bootstrap sampling can be simulated by creating K copies of each instance following a binomial distribution* [2]. When the number of instances is large, the binomial distribution tends to a Poisson distribution with $\lambda = 1$.

### 2.4 Online Boosting

Boosting is a technique that combines multiple models sequentially instead of in parallel. The main idea is that the new model is built based on the performance of the previous model, setting a higher weight to the examples that are misclassified by the previous models [12]. This achieves a better performance more generally throughout the data. Boosting was later extended for streaming and online scenarios [28, 29]. Similar to Online Bagging but where $\lambda$ of the Poisson distribution will be updated based on the correctly or incorrectly classification of the previous classifier in the ensemble. After an instance is read, the first model is trained on that instance $k$ times, where $k = Poisson(\lambda = 1)$. If the model classifies correctly that instance, the weight of that instance gets updated to a lower value. That instance is then observed by the next model, with the $\lambda$ updated to the new weighted value. If the instance is misclassified, the weight of that instance increases, so that the next model is trained more times on that instance.

## 3 RELATED WORK

Energy efficiency has been studied in computer architecture for several decades, with a focus on designing processors that can perform more operations under the same power budget [23]. In the field of machine learning, there is a current trend that focuses on reducing the energy consumption of both training and inference, to be able to run models on the edge [22]. This is the case for deep learning [6, 18, 32, 33], an area where they have also created *The Low Power Image Recognition Challenge* (LPIRC) [15], a competition where they present the best technologies that can detect objects with high precision and low energy consumption.

Data stream mining algorithms started to gain importance in the year 2000, with the publication of the Hoeffding Tree [9] algorithm. This algorithm was the first that could analyze an infinite stream of data in constant time per example. The Hoeffding Tree was then extended to handle concept drift [1, 21], that is, changes in the data source. Ensembles of Hoeffding trees were introduced to be able to achieve higher predictive accuracy. Models such as Online Bagging [28], Leveraging Bagging [4], Online Boosting [28], Online Coordinate Boosting [31], and the novel Adaptive Random Forest [17], that can also handle concept drift.

This paper focuses on the original OzaBag and OzaBoost to do a fair comparison of algorithms that were not designed to handle concept drift. While Leveraging Bagging and Adaptive Random Forest can output competitive accuracy results both in static and dynamic scenarios (with concept drift), they incur significantly higher energy requirements. The reason behind this is that more computations and memory accesses are required to be able to handle concept drift. Thus, we believe it is unfair to compare algorithms meant for concept drift scenarios with simpler algorithms under datasets without concept drift. Leveraging Bagging and Adaptive

Random Forest output significantly higher energy consumption, but a significantly higher accuracy is not observed due to the nature of the datasets.

The latest Hoeffding Tree algorithm is the already mentioned Extremely Fast Decision Tree (EFDT) (Section 2.2), which obtains higher accuracy compared to the original Hoeffding Tree while taking longer to run [26]. Concerning energy efficiency, the Vertical Hoeffding Tree (VHT) [24] algorithm was introduced as a parallel version of the Hoeffding Tree. The authors of [27] proposed a parallel version of random forests of Hoeffding trees with specific hardware configurations. GAHT, our proposed algorithm, is a hybrid between HT and EFDT, resulting in a more energy-efficient extension of EFDT and an ensemble of Hoeffding trees. This is achieved by applying the less-restrictive splitting criteria of EFDT only on a set of selected nodes and deactivating another set of less-visited nodes. More details are given in Section 4.

## 4 GREEN ACCELERATED HOEFFDING TREE

### 4.1 Basic Idea

This section presents the Green Accelerated Hoeffding Tree algorithm (GAHT), the main contribution of the paper. GAHT, presented in Alg. 1, follows a per-node splitting criterion that varies depending on the distribution of the observed data. The goal of GAHT is to present an approach that can obtain competitive accuracy results, while still staying within a constrained energy budget. In particular, our goal is to achieve the same accuracy as the EFDT and ensemble of Hoeffding trees, but at a lower energy cost.

For that, we follow similar reasoning as the authors of the *nmin adaptation* method [14]. The idea is to have dynamic hyperparameters at the nodes rather than traditional static hyperparameters. This allows the tree to split in a more personalized manner since different nodes have observed different samples of the incoming data. This approach is also useful for concept drift scenarios since it is very probable that the initial parameter setup is not valid anymore when the data has drifted.

Looking at Alg. 1, GAHT reads an instance, traverses the tree, and for each instance it calculates the fraction of instances seen at that leaf (considering the amount of instances observed since that leaf was created, and the number of leaves); as follows:

$$fraction = \frac{n_l}{n_{since\_creation}/n_{leaves}} \tag{2}$$

Where $n_l$ refers to the number of instances observed at that particular leaf, $n_{since\_creation}$ refers to the number of instances observed in the tree since that leaf was created, and $n_{leaves}$ refers to the number of leaves in the tree.

If $fraction$ is lower than the *deactivateThreshold*, that leaf is deactivated. When a leaf is deactivated the statistics are still stored, but the leaf is not allowed to grow. On the other hand, if $fraction$ is higher than the *growFastThreshold*, then that leaf is set to the EFDT splitting criteria, making faster splits on that branch. That is, instead of comparing the information gain of the two best attributes, the algorithm compares the information gain of the best attribute against the null split. The null split stands for the information gain without creating a split.

The idea behind Equation 2 is that assuming new instances arrive uniformly distributed over the leaves of the tree, the expected value

of $fraction$ is one. Hence, a value of less than one means that we have observed less than the expected amount of instances at that leaf.

We have implemented GAHT so that the user can choose the values of parameters *deactivateThreshold* and *growFastThreshold*. For these experiments we set deactivateThreshold = 0.01 and growFastThreshold = 2, to deactivate leaves observed less than 1 percent of the expected amount, and grow faster the leaves observed twice the expected amount. In the case where there are restrictions on energy consumption, the user can increase the value of *deactivateThreshold*, since deactivating more nodes will reduce energy consumption.

---

**Algorithm 1** Green Accelerated Hoeffding Tree. **Symbols:** $HT$: Initial tree; $X$: attributes; $G(\cdot)$: split evaluation function; **Hyperparameters:** $\tau$: tie threshold; $nmin$: batch size, *deactivateThreshold*: threshold to deactivate a leaf; *growFastThreshold*: threshold to set the node to follow the EFDT splitting criteria

---

1: **while** stream is not empty **do**
2:     Read instance
3:     Traverse the tree using $HT$
4:     Update statistics at leaf $l$ {Attributes}
5:     Increment $n_l$: instances seen at $l$
6:     $fraction = \dfrac{n_l}{n_{since\_creation}/n_{leaves}}$
7:     **if** $fraction <$ deactivateThreshold **then**
8:         DeactivateLeaf
9:     **else if** $fraction >$ growFastThreshold **then**
10:        growFast ← True {EFDT criteria on $l$}
11:     **end if**
12:     **if** $nmin \leq n_l$ **then**
13:        Compute $\overline{G_l}(X_i)$ for each attribute $X_i$
14:        **if** growFast == True **then**
15:           $X_b$ = null split
16:        **else**
17:           $X_b$ = second best attribute
18:        **end if**
19:        $\Delta\overline{G} = \overline{G_l}(X_a) - \overline{G_l}(X_b)$
20:        **if** $(\Delta\overline{G} > \epsilon)$ or $(\Delta\overline{G} < \epsilon < \tau)$ **then**
21:           CreateChildren($l$){New leaf $l_m$}
22:        **else**
23:           Disable att $\{X_p | (\overline{G_l}(X_p) - \overline{G_l}(X_a)) > \epsilon\}$
24:        **end if**
25:     **end if**
26: **end while**

---

### 4.2 Convergence

At any time, a leaf node in a GAHT tree exists in one of three states:

(1) If $fraction < deactivateThreshold$, the node is deactivated, meaning no further splitting is done.

(2) If the leaf node is active, and $fraction \leq growFastThreshold$, splitting of the leaf node is performed identically to a HT.

(3) If the leaf node is active, and $fraction > growFastThreshold$, splitting of the leaf node is performed identically to an EFDT.

Considering only cases (1) and (2), it is clear that the resulting tree would converge to a structure identical to that of an ordinary HT since any splitting of a leaf node is performed according to the same criteria. For case (3), proofs are provided in [25], showing that the structure of an EFDT probabilistically converges to that of a HT and, by extension, to that of an ordinary decision tree.

We note that, in an EFDT, sub-trees with internal root nodes are continuously revised using the ReEvaluateBestSplit method [25], which is not the case for a GAHT tree. This particular functionality of the EFDT algorithm exists to ensure that sub-optimal splits, resulting from eager splitting on skew attributes, are not retained within the tree structure. In GAHT, such unfortunate splits are instead pre-empted through leveraging the *nmin* parameter—since no split is performed within a leaf before *nmin* instances have been observed, the attribute skewness has a minimal effect on the tree structure (assuming that the *nmin* parameter is set to a sufficiently large value).

### 4.3 Energy complexity

In [14] the concept of energy complexity is introduced concerning Hoeffding trees. The authors conclude that the most energy-consuming part of the algorithm is related to computing the best attributes and traversing the tree. The main differences between this and the standard Hoeffding tree lie in the splitting frequency and traversing the tree.

Since the main contributor of energy consumption in a tree learning algorithm is the computation of the best splitting attributes, the energy cost of GAHT can be approximated by

$$E(GAHT) = \alpha E(S_{inactive}) + \beta E(S_{HT}) + \gamma E(S_{EFDT}), \quad (3)$$

where $E$ denotes energy cost, $S$ denotes a splitting function, and $\alpha$, $\beta$ and $\gamma$ respectively denote the proportion of leaves that exists in the deactivated state, or is being split by the HT or EFDT criteria. If $\beta = 1$, or $\gamma = 1$, the energy cost of GAHT is approximately equal to that of a HT or an EFDT, respectively. Since the deactivated leaf node state carries the least energy cost (as no splitting, and hence no computation of best splitting criteria, is performed), it is apparent that the energy cost of GAHT is upper bounded by that of HT and EFDT in the sense that

$$E(GAHT) \in O\left[sup\left\{E\left(S_{HT}\right), E\left(S_{EFDT}\right)\right\}\right]. \quad (4)$$

Similarly, we can define a lower bound of the energy cost of GAHT as

$$E(GAHT) \in \Omega\left[E\left(S_{inactive}\right)\right]. \quad (5)$$

### 4.4 Time and Space Complexity

The space and time complexity of GAHT are upper-bounded by the time and space complexity of EFDT and HT and lower bounded by deactivating leaves. The same as for energy complexity.

Regarding time complexity, following the reasoning of EFDT [26], we can divide the operations in the tree into two main ones: updating the statistics at the leaf, and evaluating the possible splits. In the case of EFDT, they re-evaluate splits, while in our case we have a simpler solution where the splits are never re-evaluated. Instead, we deactivate less promising leaves, which is more energy-efficient than re-evaluating splits.

To evaluate for possible splits, $d$ splits will be evaluated, where $d$ is the number of attributes. For each $d$ split, $v$ computations of information gains are required. Finally, each information gain requires $O(c)$ operations. Thus, split evaluations require $O(dvc)$ operations, at every leaf, every *nmin* instances. This is the case for nominal attributes; numerical attributes lead to more operations since the possible spitting candidates for each attribute need to be computed. More in-depth analysis of numerical attributes can be found in the work by [14].

Updating the statistics require to traverse the tree until the leaf, and update the table with the counts of the observed (attribute value, class) pairs. This requires $O(dvc)$, same as for HT since only the information at the leaf needs to be updated. This contrasts with the update criteria in EFDT, where the complexity is $O(hdvc)$ because more leaves are updated during the traverse process. $h$ referring to the height of the tree.

Finally, regarding space complexity, the complexity will vary depending on the input dataset and the amount of nominal and numerical attributes. For nominal attributes, a table of size $O(dvc)$ is required. Where $d$ refers to the number of attributes, $v$ the values per attribute, and $c$ the number of classes. For numerical attributes, it will depend on the method to store the statistics of each attribute. Empirical results of space complexity in the form of the size of the tree are described in Section 6.3.

## 5 EXPERIMENTAL DESIGN

### 5.1 Datasets

The algorithms have been tested on a total of 11 datasets, six synthetically generated and five real world datasets, described in Table 1. The synthetic datasets are generated with MOA [3], with the default settings. The CICIDS dataset is a cybersecurity dataset from 2017 where the task is to detect intrusions [34]. The details of the rest of the real world datasets can be found at https://moa.cms.waikato.ac.nz/datasets/. The details of the synthetic datasets can be found at the MOA official book [2] and in papers such as [5]. We omit these details due to space constraints.

### 5.2 Baselines

The goal of the experiment is to show the comparison between GAHT to the standard Hoeffding Tree algorithm; to the latest and most accurate (in regards to single online decision trees) extension of the Hoeffding Tree, i.e. EFDT; and to online decision tree ensembles. The reasoning behind comparing against an ensemble of decision trees is because we wanted to evaluate if we are able to achieve as high accuracy as for an ensemble of online decision trees, but with a single online tree, which will result in significant energy savings.

The ensembles of Hoeffding trees that are evaluated are Oza-aBag [28] (Online Bagging) and OzaBoost [28] (Online Boosting), to have a fair comparison with algorithms that are not designed specifically for concept drift, such as HT, GAHT, and EFDT.

**Table 1: Synthetic and real world datasets. $A_i$ and $A_f$ are the number of nominal and numerical attributes, respectively. Abbrv. is the abbreviation used for Table 3**

| Dataset | Instances | Abbrv. | $A_i$ | $A_f$ | Classes |
|---|---|---|---|---|---|
| **Synthetic** | | | | | |
| RandomTree | 1,000,000 | RTRee | 5 | 5 | 2 |
| Waveform | 1,000,000 | Wave | 0 | 21 | 3 |
| RandomRBF | 1,000,000 | RBF | 0 | 10 | 2 |
| LED | 1,000,000 | LED | 24 | 0 | 10 |
| Hyperplane | 1,000,000 | Hyper | 0 | 10 | 2 |
| Agrawal | 1,000,000 | Agrawal | 3 | 6 | 2 |
| **Real World** | | | | | |
| Airline | 539,383 | Airline | 4 | 3 | 2 |
| Poker | 829,201 | Poker | 5 | 5 | 10 |
| CICIDS | 461,802 | CICIDS | 78 | 5 | 6 |
| Forest | 581,012 | Forest | 40 | 10 | 7 |
| Electricity | 45,312 | Elec | 1 | 6 | 2 |

## 5.3 Environment–Setup

All algorithms are run in MOA. For the EFDT, we have used the implementation in MOA provided by the original authors[1]. Our algorithm, GAHT, has been coded and tested in MOA (Section 5.5). We set `deactivateThreshold=0.01` and `growFastThreshold=2` for GAHT, after having tested different parameter setups. Thus, we deactivate all leaves that have been observed less than 1 percent of the expected observations; splitting faster on the leaves that observed at least twice as many instances as the expected amount. The accuracy was evaluated using a *test-then-train* approach, where each instance is first used for testing and then for training. The performance is calculated every 100k instances for all datasets, except every 4k for the *Electricity* dataset.

## 5.4 Metrics

The following metrics are evaluated: accuracy, total energy consumption, and size of the tree. The accuracy is measured as the percentage of correctly classified instances.

The total energy is measured in joules, as the energy consumed by the processor plus the energy consumed by the DRAM while executing the algorithm. The energy is measured using Intel's RAPL interface [7] which is integrated into Intel Power Gadget[2], a tool developed by Intel that outputs power and energy real-time measurements. This interface accesses specific hardware counters available in the processor, which saves energy-related measurements of the processor and the DRAM. Since Intel Power Gadget does not isolate the energy consumption of a specific program, we have ensured that only our MOA experiments were running on the machine at the time of the measurements. All experiments were run five times and averaged the results. All the experiments are run on a machine with a 3.5 GHz Intel Core i7, with 16GB of RAM, running macOS.

[1]https://github.com/chaitanya-m/kdd2018
[2]https://software.intel.com/en-us/articles/intel-power-gadget-20

## 5.5 Reproducibility

Aiming for reproducibility, the code of GAHT (as a part of MOA) is available in the following link: https://www.dropbox.com/sh/a5rozyjvbizsctu/AAC1pISlO3aD7F9HoRyKCK4ba?dl=0. It will be publicly available in GitHub once the paper has been reviewed, but we only share the Dropbox link in this manuscript to comply with the double-blind review policy.

## 6 EMPIRICAL EVALUATION

Table 3 shows the accuracy, energy consumption, and size of the tree results of running GAHT, EFDT, HT, OzaBag, and OzaBoost on the 11 different datasets. The average results comparing GAHT against the other algorithms in terms of energy consumption and accuracy are presented in Table 2. We detail the results about each specific metric on the following sections (Sections 6.1,6.2, and 6.3).

**Table 2: Difference in accuracy ($\Delta$A) and energy consumption ($\Delta$E). The difference is measured as the percentage reduction between the algorithms. A positive value in accuracy means that GAHT (our proposed solution) achieves higher accuracy than the compared algorithm. A negative value in energy means that GAHT was able to reduce the energy consumption by that percent.**

| Dataset | GAHT vs EFDT | | GAHT vs HT | | GAHT vs OzaBag | | GAHT vs OzaBoost | |
|---|---|---|---|---|---|---|---|---|
| | $\Delta$A (%) | $\Delta$E(%) | $\Delta$A (%) | $\Delta$E(%) | $\Delta$A(%) | $\Delta$E (%) | $\Delta$A (%) | $\Delta$E(%) |
| RTree | 0.06 | -26.28 | 1.73 | 22.29 | 1.15 | -78.12 | -0.46 | -84.91 |
| Wave | 1.53 | -24.75 | -0.62 | 151.48 | -2.24 | -58.51 | -0.77 | -62.46 |
| RBF | 1.12 | -38.64 | 1.38 | 40.95 | -1.28 | -73.49 | -0.78 | -75.34 |
| LED | 0.11 | -41.69 | -0.32 | 62.02 | 0.06 | -71.99 | 0.15 | -73.72 |
| Hyper | 1.55 | -29.27 | -1.65 | 59.30 | -1.80 | -69.49 | -1.54 | -71.15 |
| Agrawal | -0.11 | -25.04 | 0.34 | 32.81 | -0.57 | -69.16 | 1.38 | -84.12 |
| Airline | -0.38 | -20.49 | -0.37 | 36.53 | -0.13 | -80.95 | -0.28 | -81.63 |
| Poker | 1.31 | -21.67 | 8.90 | 33.82 | 5.24 | -64.39 | -1.80 | -70.33 |
| CICIDS | 0.08 | -28.67 | 0.10 | -4.51 | 0.18 | -66.08 | -0.10 | -70.14 |
| Forest | -3.67 | -32.28 | 9.18 | 33.59 | 1.77 | -64.06 | -4.42 | -72.39 |
| Elec | 0.14 | -11.35 | 2.89 | 22.46 | 1.78 | -40.41 | -4.13 | -42.09 |
| Average | 0.16 | -27.29 | 1.96 | 44.61 | 0.38 | -66.97 | -1.16 | -71.66 |

## 6.1 Accuracy

Figure 1 shows a comparison between the accuracy consumed by the different algorithms on the different datasets. We can observe how GAHT outputs higher accuracy than HT, EFDT, and OzaBag on the majority of datasets. On average, GAHT achieves 0.16% more accuracy than EFDT, 0.38% more than OzaBag, 2% more than HT, and 1.16% less than OzaBoost (Table 2).

Table 4 ranks the algorithms based on their accuracy and energy consumption. We observe that GAHT obtains an average rank of 2.91, compared to 2.27 from OzaBoost, the one with the highest score. The other algorithms have either lower rank (EFDT and HT) or the same rank (OzaBag). A Friedman test [13] does not indicate any significant differences in accuracy between the algorithms ($\alpha = 0.05$), suggesting that GAHT achieves an average accuracy comparable to that of each of the other algorithms tested.

**Table 3: Accuracy, energy consumption and tree size results between the GAHT, EFDT, HT, OzaBag(OBag), and Oza-Boost(OBoost). Number of inactive leaves (In.Leaves) and number of fast nodes (F.Nodes) are presented for GAHT. Fast nodes are GAHT nodes with the EFDT splitting criteria**

| | Accuracy (%) | | | | | Total Energy(J) | | | | | Nodes | | | | | In.leaves | F.Nodes |
|---------|-------|-------|-------|-------|--------|--------|--------|--------|---------|---------|----------|----------|---------|----------|----------|-------|-------|
| Dataset | GAHT | EFDT | HT | OBag | OBoost | GAHT | EFDT | HT | OBag | OBoost | GAHT | EFDT | HT | OBag | OBoost | GAHT | GAHT |
| RTree | 96.74 | 96.68 | 95.01 | 95.59 | 97.20 | 161.02 | 218.42 | 131.67 | 735.88 | 1066.85 | 1988.00 | 2572.00 | 1132.00 | 11459.00 | 17988.00 | 0.00 | 802.00 |
| Wave | 83.60 | 82.07 | 84.22 | 85.84 | 84.37 | 444.87 | 591.17 | 176.90 | 1072.12 | 1185.15 | 5195.00 | 4611.00 | 323.00 | 3312.00 | 3958.00 | 4.00 | 3937.00 |
| RBF | 93.31 | 92.19 | 91.93 | 94.59 | 94.09 | 205.14 | 334.32 | 145.54 | 773.82 | 831.79 | 3037.00 | 3567.00 | 819.00 | 8092.00 | 9478.00 | 7.00 | 1915.00 |
| LED | 74.18 | 74.07 | 74.50 | 74.12 | 74.03 | 260.79 | 447.25 | 160.96 | 931.16 | 992.50 | 1669.00 | 2795.00 | 107.00 | 1004.00 | 1010.00 | 0.00 | 1068.00 |
| Hyper | 88.49 | 86.94 | 90.14 | 90.29 | 90.03 | 210.96 | 298.26 | 132.43 | 691.34 | 731.35 | 3191.00 | 3513.00 | 665.00 | 6868.00 | 6978.00 | 0.00 | 2325.00 |
| Agrawal | 94.44 | 94.55 | 94.10 | 95.01 | 93.06 | 126.81 | 169.17 | 95.49 | 411.26 | 798.69 | 775.00 | 1713.00 | 408.00 | 3318.00 | 12043.00 | 4.00 | 264.00 |
| Airline | 60.30 | 60.68 | 60.67 | 60.43 | 60.58 | 227.81 | 286.52 | 166.86 | 1195.68 | 1240.31 | 19755.00 | 30917.00 | 8603.00 | 84805.00 | 88553.00 | 0.00 | 6833.00 |
| Poker | 90.67 | 89.36 | 81.77 | 85.43 | 92.47 | 208.55 | 266.26 | 155.84 | 585.62 | 702.95 | 4700.00 | 2240.00 | 903.00 | 2978.00 | 2993.00 | 169.00 | 3806.00 |
| CICIDS | 99.76 | 99.68 | 99.66 | 99.58 | 99.86 | 296.63 | 415.87 | 310.65 | 874.51 | 993.43 | 129.00 | 161.00 | 63.00 | 614.00 | 880.00 | 14.00 | 83.00 |
| Forest | 86.80 | 90.47 | 77.62 | 85.03 | 91.22 | 410.37 | 605.96 | 307.20 | 1141.89 | 1486.54 | 3147.00 | 2200.00 | 699.00 | 3240.00 | 2946.00 | 53.00 | 2709.00 |
| Elec | 82.37 | 82.23 | 79.48 | 80.59 | 86.50 | 58.59 | 66.09 | 47.84 | 98.31 | 101.18 | 341.00 | 270.00 | 77.00 | 684.00 | 700.00 | 0.00 | 255.00 |

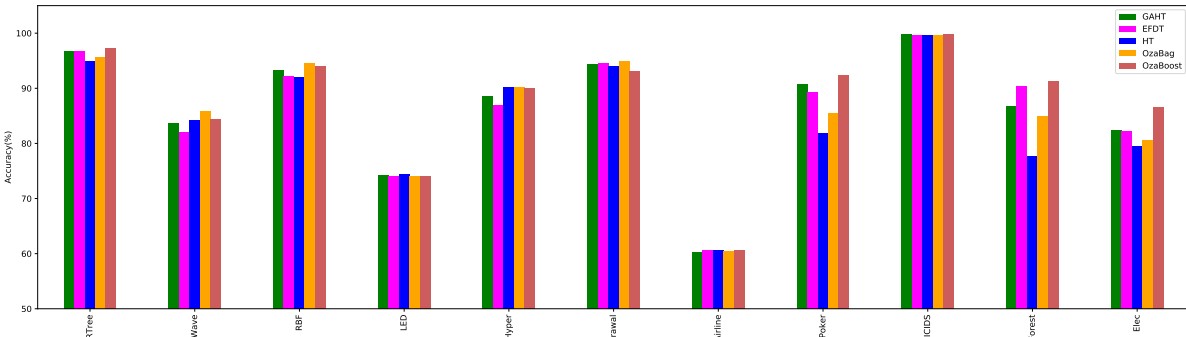

**Figure 1: Accuracy comparison between GAHT, EFDT, HT, OzaBag, and OzaBoost, measured as the percentage of correctly classified instances**

## 6.2 Energy Consumption

The comparison of energy consumption between the different algorithms is shown in Figure 2. We can observe how GAHT is consuming less energy than all algorithms except for HT, matching with the theoretical assumptions from Section 4.3, where we stated that the energy consumption of GAHT is upper bounded by the energy consumption of HT and EFDT. From Table 2 we observe that, on average, GHAT consumes 67% less energy than OzaBag, 27% less energy than EFDT, and 71% less energy than OzaBoost, while still attaining comparable or better accuracy on average.

Figure 2 shows how well the algorithms scale with the number of instances. While the single decision trees (GAHT, HT, EFDT) are able to scale well with the number of instances, the ensemble models (OzaBag, OzaBoost) produce a significantly steeper curve.

The energy rankings (Table 4) are more uniform compared to the accuracy rankings. We can see, e.g., that GAHT is always ranking second (except for one dataset). A Friedman test [13] followed by a Bonferroni-Dunn post-hoc test [10, 11] indicates significant

differences ($\alpha = 0.05$) between all pairs of algorithms with respect to energy consumption, except for GAHT vs. HT. That is, GAHT consumes significantly less energy than all other tested algorithms except for HT—the algorithm with the worst average rank with respect to accuracy. OzaBoost is consistently ranked last, consuming significantly more energy than all other tested algorithms, showcasing the dilemma of how to choose the most appropriate solution. The ideal scenario is to choose an algorithm that has a lower energy rank and a lower accuracy rank. This relationship between accuracy and energy is portrayed in Figure 3. While HT is ranked first in energy consumption, it is also ranked last in terms of accuracy. The opposite occurs for OzaBoost, which is ranked first in accuracy and last in energy consumption. GAHT, on the other hand, is second in accuracy and second in energy consumption, which seems the most balanced option of all.

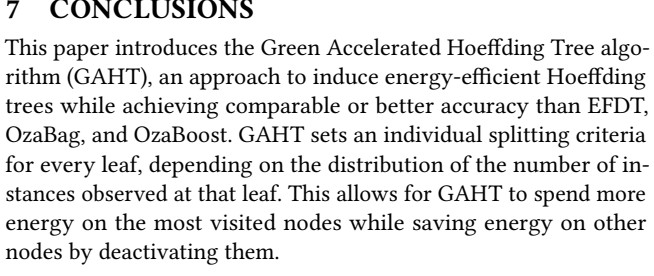

**Figure 2: Energy consumption increase with the number of instances. The figure shows how single online decision trees (GAHT, HT, EFDT) are able to scale better than ensembles of trees (OzaBag and OzaBoost) when increasing the number of instances**

## 6.3 Memory Footprint

GAHT is able to create smaller trees than the other algorithms (except for HT) in the majority of datasets. There are some datasets (e.g. poker dataset) where GAHT is creating a higher number of trees, due to the characteristics of the data. However, even though GAHT is outputting bigger trees for this particular dataset, it is still able to achieve between 65% and 70% less energy than the ensemble algorithms. The reason is that not as many trees need to be traversed, and fewer memory accesses need to be carried out.

These models were still in the range from 1MB to 30MB, so fairly small compared to deep learning algorithms, which can take more than 1GB of space [8].

## 7 CONCLUSIONS

This paper introduces the Green Accelerated Hoeffding Tree algorithm (GAHT), an approach to induce energy-efficient Hoeffding trees while achieving comparable or better accuracy than EFDT, OzaBag, and OzaBoost. GAHT sets an individual splitting criteria for every leaf, depending on the distribution of the number of instances observed at that leaf. This allows for GAHT to spend more energy on the most visited nodes while saving energy on other nodes by deactivating them.

**Table 4: Algorithm ranking between energy consumption and accuracy. The rank is measured as the algorithm that scored the highest accuracy or the lowest energy. The lower the better.**

|  | Accuracy | | | | | Energy | | | | |
|---|---|---|---|---|---|---|---|---|---|---|
|  | GAHT | EFDT | HT | OzaBag | OzaBoost | GAHT | EFDT | HT | OzaBag | OzaBoost |
| RTree | 2.00 | 3.00 | 5.00 | 4.00 | 1.00 | 2.00 | 3.00 | 1.00 | 4.00 | 5.00 |
| Wave | 4.00 | 5.00 | 3.00 | 1.00 | 2.00 | 2.00 | 3.00 | 1.00 | 4.00 | 5.00 |
| RBF | 3.00 | 4.00 | 5.00 | 1.00 | 2.00 | 2.00 | 3.00 | 1.00 | 4.00 | 5.00 |
| LED | 2.00 | 4.00 | 1.00 | 3.00 | 5.00 | 2.00 | 3.00 | 1.00 | 4.00 | 5.00 |
| Hyper | 4.00 | 5.00 | 2.00 | 1.00 | 3.00 | 2.00 | 3.00 | 1.00 | 4.00 | 5.00 |
| Agrawal | 3.00 | 2.00 | 4.00 | 1.00 | 5.00 | 2.00 | 3.00 | 1.00 | 4.00 | 5.00 |
| Airline | 5.00 | 1.00 | 2.00 | 4.00 | 3.00 | 2.00 | 3.00 | 1.00 | 4.00 | 5.00 |
| Poker | 2.00 | 3.00 | 5.00 | 4.00 | 1.00 | 2.00 | 3.00 | 1.00 | 4.00 | 5.00 |
| CICIDS | 2.00 | 3.00 | 4.00 | 5.00 | 1.00 | 1.00 | 3.00 | 2.00 | 4.00 | 5.00 |
| Forest | 3.00 | 2.00 | 5.00 | 4.00 | 1.00 | 2.00 | 3.00 | 1.00 | 4.00 | 5.00 |
| Elec | 2.00 | 3.00 | 5.00 | 4.00 | 1.00 | 2.00 | 3.00 | 1.00 | 4.00 | 5.00 |
| Average | 2.91 | 3.18 | 3.73 | 2.91 | 2.27 | 1.91 | 3.00 | 1.09 | 4.00 | 5.00 |

We have evaluated the performance in terms of energy consumption and accuracy between GAHT, EFDT, HT, OzaBag, and

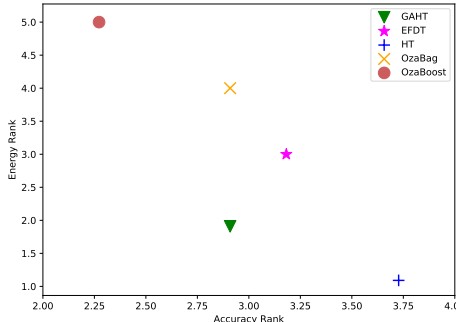

**Figure 3: Relationship between the energy ranks ($y$ axis) and accuracy ranks($x$ axis). A low rank in energy and a low rank in accuracy is ideal, which would mean that the lowest energy consuming algorithm is also the most accurate**

OzaBoost on 11 datasets. The results show that GAHT consumes significantly less energy than EFDT (27% on average), OzaBag (67% on average), and OzaBoost (72% on average) while still attaining comparable or better accuracy on average. GAHT consumes more energy than HT—the algorithm with the worst average rank with respect to accuracy. In terms of the memory footprint, GAHT outputs smaller models than EFDT, OzaBag, and OzaBoost for the majority of datasets. Even bigger GAHT trees would output lower energy consumption than a smaller ensemble of HT trees. Our models take between 1MB to 30MB of space, which is a fairly small amount compared to many deep learning solutions.

The results of this paper show that it is possible to create more energy-efficient data stream mining algorithms while maintaining accuracy levels of ensembles of Hoeffding trees. This contributes to moving stream mining algorithms to the edge, by smartly using the energy only on the necessary parts of the algorithm.

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
