# OpenReview forum: "Green Accelerated Hoeffding Tree"
_tinyml.org/tinyML/2021/Research_Symposium — tinyML 2021 Poster_

### Official Review · AnonReviewer2 · 2021-01-29

**Overall Merit Score:** 2

**Brief Summary:**

This paper proposes an energy efficient method, Green Accelerated Hoeffding Tree, for mining infinite streams of data. The main idea of the paper is to split a leaf in the tree via different criteria (e.g. HT / EFDT or directly deactivate) when the fraction of instances seen at the leaf varys. Overall, the proposed GAHT achieves better accuracy and energy comparing with EFDT, online bagging algorithms and higher accuracy comparing with HT.

**Detailed Comments:**

Please see the "paper strengths" and "paper weaknesses" for details

**Paper Strengths:**

(1) The writing is clear and easy to understand.

(2) The proposed method achieves good energy reduction over existing methods, while keeping the accuracy.

**Paper Weaknesses:**

(1) Regarding novelty, it seems that the paper directly combines HT and EFDT with handcrafted routing criterion (the fraction), which doesn't seem to be quite impressive.

(2) The proposed method is deployed on desktop CPUs, which is still much more powerful comparing with TinyML devices such as the Raspberry Pi or even Microcontrollers (which has <2M read-only memory and <512K read-write memory). For a TinyML submission, I doubt whether the proposed method can be deployed on these tiny devices (espcially Microcontrollers, on which deep learning models can actually be deployed [1-3]) given the relatively large model size (up to 30MB).

(3) It will be better to present more detailed analysis such as latency improvement, DRAM usage improvement, and power improvement to understand where the energy reduction comes from.


[1] Saha et al., RNNPool: Efficient Non-linear Pooling for RAM Constrained Inference. In NeurIPS 2020.

[2] Lin et al., MCUNet: Tiny Deep Learning on IoT Devices. In NeurIPS 2020.

[3] Memory-Driven Mixed Low Precision Quantization For Enabling Deep Network Inference On Microcontrollers. In SysML 2020.


**Poster (If Paper Is Rejected):**

1: Yes, ok for poster sesion to nurture work

**Reviewer Confidence:**

3: The reviewer is fairly confident that the evaluation is correct

---

### Official Review · AnonReviewer3 · 2021-01-31

**Overall Merit Score:** 2

**Brief Summary:**

This paper presents a new decision tree algorithm that provides an improved trade-off between accuracy and efficiency relative to the existing standard Hoeffding tree and Extremely Fast Hoeffding Tree algorithms.


**Detailed Comments:**


* Intro: Please describe the type of applications where this algorithm would be appropriate and give a couple of examples.  It would be especially good to have an example that fits the "Tiny" use case (i.e. a case where power and size constraints are significant).

* Typo, Pg 1 line 110 "Related works to our are presented" => "Related works are presented"

* Typo, Pg 1 line 112" "together with converge" -> "together with convergence"

* Please provide a brief description of the data stream mining problem. What limitations are necessary to make the following statement true? "This [Hoeffding Tree] algorithm  was the first that could analyze an infinite stream of data in constant time per example."  For example, could online K-means be considered a data stream mining algorithm?  It operates in constant time per example.

* Section 2.1, define G, R, and tau.

* Typo line 221, "that can also handle" => "can also handle"

* Section 4.1  After a leaf is deactivated, are instances still assigned to that leaf, or can it just not be split?

* Section 5.  Some basic description of the datasets is needed.  A limited explanation can be justified by the space constraints, but without any description, the reader will have difficulty understanding the experiments.

* Section 6.1.  Can you suggest an explanation for why GAHT achieves higher accuracy than EFDT?  From the earlier explanation, it seemed that GAHT was something of an interpolation between HT and EFDT, so one would expect accuracy between those two as well.

* Over the 11 different test sets, are any trends apparent that suggest that certain algorithms perform better on problems with certain characteristics?

* Table 4: Averaging the rank can either emphasize small differences or minimize large differences.  A better way to aggregate results over the several tests might be to normalize performance on each task; for example average the energy as a fraction of the most energy-expensive algorithm for each task, or average the accuracy loss relative to the most accurate algorithm on each task.   As it is, it is difficult to tell how significant the differences in accuracy and energy are, so there is no real guidance for an engineer asking questions like "Can I afford the extra power relative to an HT?" or "Is the accuracy penalty relative to OzaBoost tolerable?".



**Paper Strengths:**

* The authors present a detailed comparison with other algorithms.
* The paper provides an intuitive explanation for why the proposed algorithm is expected to provide better accuracy than the standard HT algorithm and better efficiency than the EFDT.


**Paper Weaknesses:**

* The paper presumes familiarity with existing art in decision tree algorithms, so may be difficult to follow for readers who are not very familiar with that area.

* It is not clear what real-world problems would be good candidates for this algorithm.


**Poster (If Paper Is Rejected):**

1: Yes, ok for poster sesion to nurture work

**Reviewer Confidence:**

3: The reviewer is fairly confident that the evaluation is correct

---

### Decision · Program_Chairs · 2021-02-05

**Decision:**

Accept (Poster)

**Comment:**

Based on the reviewer feedback, your paper has been accepted as a poster.

Please read the reviews carefully and make sure the concerns are addressed in your poster submission.

Accepted posters are given a 5-minute slot for an oral presentation on Friday, March 26, 2021, to pitch the main ideas of your work and to stimulate discussions. Detailed instructions will follow soon. All final posters will earn a stamp of acceptance as such: “Published as a poster at TinyML Research Symposium 2021.”